# Unsupervised Federated Domain Adaptation for Segmentation of MRI Images

## Abstract

Automatic semantic segmentation of magnetic resonance imaging (MRI) images using deep neural networks greatly assists in evaluating and planning treatments for various clinical applications. However, training these models is conditioned on the availability of abundant annotated data to implement the supervised learning procedure. Even if we annotate enough data, MRI images display considerable variability due to factors such as differences in patients, MRI scanners, and imaging protocols. This variability necessitates retraining neural networks for each specific application domain, which, in turn, requires manual annotation by expert radiologists for all new domains. To relax the need for persistent data annotation, we develop a method for unsupervised federated domain adaptation using multiple annotated source domains. Our approach enables the transfer of knowledge from several annotated source domains to adapt a model for use in an unannotated target domain. Initially, we ensure that the target domain data shares similar representations with each source domain in a latent embedding space, modeled as the output of a deep encoder, by minimizing the pair-wise distances of the distributions for the target domain and the source domains. We then employ an ensemble approach to leverage the knowledge obtained from all domains. We perform experiments on two datasets to demonstrate our method is effective. Our code is available as a supplement: `http//:SupressedforDoubleBlindReview`.

## 1 Introduction

Semantic segmentation of MRI images can help to detect anatomical structures or regions of interest in these images to simplify the interpretation of these images. High-quality segmented images are extremely useful in applications such as disease detection and monitoring Hatamizadeh et al. (2021); Karayegen & Aksahin (2021), surgical Guidance Jolesz et al. (2001); Wei et al. (2022), treatment response assessment Kickingereder et al. (2019), and AI-aided diagnosis Arsalan et al. (2019). UNet-based convolutional neural networks (CNNs) have shown to be effective for automatic semantic segmentation of MRI images Pravitasari et al. (2020); Maji et al. (2022), but their adoption in clinical settings has been quite limited. A major reason for this limitation is that training deep neural networks requires large annotated datasets Yang et al. (2021); Wang et al. (2021). Annotating MRI data reliably requires the expertise of trained radiologists and physicians which makes it a challenging process. Moreover, using crowdsourcing annotation platforms would be inapplicable because medical data is normally distributed in different institutions and due to the lack of specialized knowledge by an average person. Moreover, medical data is private and sharing it for annotation is highly regulated.

Even if we prepare a suitable annotated datasets and successfully train a segmentation model, it may not generalize well in practice. The reason is that MRI images are known to be significantly variable due to differences in patients, MRI scanners, and imaging protocols Kruggel et al. (2010); Ackaouy et al. (2020). These variances introduce domain shift during testing Sankaranarayanan et al. (2018b) which leads to model performance degradation Xu et al. (2019). Annotating data persistently and then retraining the model from scratch may address this challenge but is an inefficient solution. Unsupervised Domain Adaptation (UDA) is a framework that has been developed to tackle the issue of domain shift without requiring data annotation. The goal in UDA is to enable the generalization of a model which is trained on a source domain with annotated data to a target domain with only unannotated data Biasetton et al. (2019); Zou et al. (2018).

A major approach to address UDA is to map data points from a source and a target domain into a shared latent embedding space at which the two distributions are aligned. Since domain-shift would not exist in such a latent feature space, a segmentation model which is trained solely using the source domain data and receives latent features at its input, would generalize on the target domain data. The major approach to implement this idea is to model the data mapping function using a deep neural encoder network, where its output-space models the shared latent space. The encoder is trained such that it aligns the source and the target distributions at its output. This process can be achieved using adversarial learning (Javanmardi & Tasdizen (2018); Cui et al. (2021); Sun et al. (2022)) or direct probability matching (Bhushan Damodaran et al. (2018); Ackaouy et al. (2020); Al Chanti & Mateus (2021)). In the former approach, the distributions are matched indirectly through competing generator and discriminator networks to learn a domain-agnostic embedding at the output of the generator. In the latter approach, a probability metric is selected and minimized to align the distributions directly in the latent embedding space.

Most UDA methods utilize a single source domain for knowledge transfer. However, we may have access to several source domains. Specifically, medical data is usually distributed in different institutions and often we can find several source domains. For this reason, classic UDA has been extended to multi-source UDA (MSUDA), where the goal is to benefit from multiple distinct sources of knowledge Zhao et al. (2019); Tasar et al. (2020); Gong et al. (2021); He et al. (2021). The possibility of leveraging collective information from multiple annotated source domains can enhance model generalization compared to single-source UDA. Unlike single-source UDA, MSUDA algorithms need to consider the differences in data distribution between pairs of source domains in addition to the disparities between a single source domain and the target domain.

A naive approach to address MSUDA is to assume that the annotated source datasets can be transferred to a central server and then processed similar to single-source UDA. However, this assumption overlooks potential common constraints in medical domain problems such as privacy and security regulations. These regulations often prevent sharing data across the source domains. To overcome these challenges, our contribution is an alternative two-step MSUDA algorithm: (1) we first train a model based single-source UDA between each source domain and the target domain. We rely on direct probability metric minimization for this purpose. (2) During the testing time on the target domain, we use these models individually to segment an image and then aggregate the resulting segmented images according to the confidence we have in each model in a pixel-wise manner. As a result, we maintain the privacy constraints between the source domains and improve upon single-source UDA algorithms. Moreover, we implemented ensemble multi-source UDA techniques on the MICCAI 2016 and 2019 CHAOS MR datasets to show that our two-step MSUDA algorithm is robust across these medical image segmentation datasets and leads to state-of-the-art performance.

## 2 Related work

**Semantic Segmentation of MRI Data** Semantic segmentation of MRI images helps to increase the clarity and interpretability of these images Işın et al. (2016). While this task is often performed manually by radiologists in clinical settings, manual annotations is prone to inter-reader variations, expensive, and time-consuming. To address these limitations, classical machine learning algorithms have been used to automate segmenting MRI scans Levinski et al. (2009); Liu & Guo (2015); Carreira et al. (2012); Sourin et al. (2010). However, these algorithms rely on hand-crafted features which require expertise in engineering and medicine, and careful creation of imaging features given a specific problem of interest. Additionally, anatomical variations, variations in MRI acquisition settings and scanners, imperfections in image acquisition, and variations in pathology appearance serve as obstacles for their generalization in clinical settings.

Deep learning models have the capacity to relax the need for feature engineering. Specifically, architectures based on convolutional neural networks (CNNs) have been found quite effective in medical semantic segmentation Long et al. (2015a); Ronneberger et al. (2015a); Du et al. (2020). Fully Convolutional Networks (FCNs) Du et al. (2020) extend the vanilla CNN architecture to an end-to-end model for pixel-wise prediction which is more suitable for semantic segmentation. FCNs have an encoder-decoder structure, where the core idea is to replace fully connected layers of a CNN with up-sampling layers that map back the features that are extracted by the convolutional layers to the original input space dimension. This way, the model can be trained to predict the semantic masks directly at its output. As an extension to FCNs, U-Nets Ronneberger

et al. (2015a) are the dominant architecture for medical semantic segmentation tasks. U-Nets are similar to FCNs, but skip connections between the encoder and decoder layers are used to preserve spatial information at all abstraction levels. For this reason, the number of down-sampling and up-sampling layers are equal in a U-Net to make adding skip connections between pairs of layers that have the same hierarchy possible. Similar to CNNs, skip connections help propagating the spatial information is in deeper layers of U-Nets which helps to have accurate segmentation results through using features with different abstraction levels. The downside of U-Nets is the necessity of having large annotated datasets to train them.

**Single-Source UDA** are developed to relax the need for persistent data annotation and improve model generalization using solely unannotated data. These methods utilize only one source domain with annotated data to adapt a model to generalize on the unannotated target domain. The notion of domain is subjective and can be even defined for the same problem if a condition is changed during the model testing phase. UDA methods have been used extensively on the two areas of image classification Goodfellow et al. (2014); Hoffman et al. (2018); Dhouib et al. (2020); Luc et al. (2016); Tzeng et al. (2017); Sankaranarayanan et al. (2018a); Long et al. (2015b; 2017); Morerio et al. (2018) and image segmentation Javanmardi & Tasdizen (2018); Cui et al. (2021); Sun et al. (2022); Bhushan Damodaran et al. (2018); Ackaouy et al. (2020); Al Chanti & Mateus (2021). The classic workflow in UDA is to train a deep neural network on both the annotated source domain and the unannotated target domain such that the end-to-end learning is supervised by the source domain data and domain alignment is realized in a network hidden layer as a latent embedding space using data from both domains. As a result, the network would generalize on the target domain. The alignment of the distributions for UDA is often achieved by utilizing generative adversarial networks Goodfellow et al. (2014); Hoffman et al. (2018); Dhouib et al. (2020); Javanmardi & Tasdizen (2018); Cui et al. (2021); Sun et al. (2022) or probability metric minimization Long et al. (2015b; 2017); Morerio et al. (2018); Bhushan Damodaran et al. (2018); Ackaouy et al. (2020); Al Chanti & Mateus (2021). Adversarial learning aligns two distributions indirectly at the output of the generative subnetwork. For metric minimization, we minimize a suitable probability metric between the embeddings of the source and target domains Long et al. (2015b; 2017); Morerio et al. (2018); Bhushan Damodaran et al. (2018); Ackaouy et al. (2020); Al Chanti & Mateus (2021); Rostami et al. (2020) and minimize it at the output of a shared encoder for direct distribution alignment. The upside of this approach is that it requires less hyperparameter tuning. However, single-Source UDA algorithms do not leverage inter-domain statistics when multiple source domains are present. Therefore, extending single-source UDA algorithms to a multi-source federated setting is a non-trivial task that requires careful consideration to mitigate the negative effect of distribution mismatches between several source domains.

**Multi-Source UDA** is an extension to single-source UDA to benefit from multiple source domains to enhance the model generalization on a single target domain Xu et al. (2018); Zhao et al. (2019); Tasar et al. (2020); Gong et al. (2021). MSUDA is a more challenging problem due to variances across the source domains. Xu et al. 2018 extended adversarial learning to MSUDA by first reducing the difference between source and target domains using multi-way adversarial learning and then integrating the corresponding category classifiers. Zhao et al. Zhao et al. (2019) extend this idea by introducing dynamic semantic consistency in addition to using the pixel-level cycle-consistently towards the target domain. StandardGAN Tasar et al. (2020) relies on adversarial learning but it standardizes data for each source and target domains so that all domains share similar distributions to reduce the adverse effect of variances. Peng et al. 2019a align inter-domain statistics of source domains in an embedding space to mitigate the effect of domain shift between the source domains. Guo et al. 2018 adopt a meta-learning approach to combine domain-specific predictions, while Venkat et al. Venkat et al. (2020) use pseudo-labels to improve domain alignment. Note that having more source data in the MUDA setting does not necessarily lead to improved performance compared to single-source UDA because negative transfer, where adaptation from one domain hinders performance in another, can degrade the performance compared to using single-source UDA. Li et al. 2018 leverage domain similarity to avoid negative transfer by utilizing model statistics in a shared embedding space. Zhu et al. 2019 align deep networks at different abstraction levels to achieve domain alignment. Wen et al. 2020 introduce a discriminator to exclude data samples with a negative impact on generalization. Zhao et al. 2020 align target features with source-trained features using optimal transport and combine source domains proportionally based on the optimal transport distance. mDALUA Gong et al. (2021) address the effect of negative transfer using domain attention, uncertainty maximization, and attention-guided adversarial alignment.

**Federated Learning (FL)** is a distributed learning technique where the goal is to train distinct models in a decentralized manner when the data is distributed. Instead of the naive solution of transmitting raw data, the prediction by models are fused to preserve data privacy. The core idea in FL is to learn a model using the local data while not sharing data and then use the individual model predictions to come up with an improved collective performance. As a result, data privacy is perserved and we can benefit from distributed computational resources. While works on FL for supervised learning are extensive Bai et al. (2021); Niu & Deng (2022); Wicaksana et al. (2022), its application in UDA remains mostly unexplored. There are several FL studies on UDA, but these methods are primarily designed for classification tasks Peng et al. (2019b); Song et al. (2020). In our work, we develop a federated multi-source UDA algorithm.

## 3 Problem Formulation

Our focus in this work is to train a segmentation model for a target domain with the data distribution $\mathcal{T}$, where only unannotated images are accessible, i.e., we observe unannotated samples $\mathcal{D}^T = \{\boldsymbol{x}_1^t, \ldots, \boldsymbol{x}_{n^t}^t\}$ from the target domain distribution $\mathcal{T}$. Each data point $\boldsymbol{x} \in \mathbb{R}^{W \times H \times C}$ in the input space is an $W \times H \times C$ MRI image, where $W, H,$ and $C$ denote the width, height, and the number of channels for the image. The goal is to segment an input image into semantic classes which are clinically meaningful, e.g., different organs in a frame. Since training a segmentation model with unannotated images is an ill-posed problem, we consider that we also have access to $N$ distinct domains with the data distributions $\mathcal{S}_1, \mathcal{S}_2 \ldots \mathcal{S}_N$, where annotated segmented images are accessible in each domain, i.e., we have access to the annotated samples $\mathcal{D}_k^S = \{(\boldsymbol{x}_{k,1}^s, \boldsymbol{y}_{k,1}^s), \ldots, (\boldsymbol{x}_{k,n_k^s}^s, \boldsymbol{y}_{k,n_k^s}^s)\}$, where $\boldsymbol{x}_k^s \sim \mathcal{S}_k$ and $\forall i, j : \mathcal{S}_i \neq \mathcal{S}_j, \mathcal{S}_i \neq \mathcal{T}$. Each point $\boldsymbol{y}$ in the output space is a semantic mask with the same size of the input MRI images, prepared by a medical professional. We consider a segmentation model $f_\theta(\cdot) : \mathbb{R}^{W \times H \times C} \to \mathbb{R}^{|\mathcal{Y}|}$ with learnable parameters $\theta$, e.g., 3D U-Net Ahmad et al. (2021), that should be trained to map the input image into a semantic mask, where $|\mathcal{Y}|$ is the number of shared semantic classes across the domains, determined by clinicians according to a specific problem. It is crucial to note that the semantic classes are the same classes across all the domains.

To train a generalizable segmentation model with a single source domain, we can rely on the common approach of UDA, where we adapt a source-trained model to generalize better on the target domain. To this end, we can first train the segmentation model for the single source domain. This is a straightforward task which can be performed using empirical risk minimization (ERM) on the corresponding annotated dataset:

$$\theta_k = \arg\min_\theta \mathcal{L}_{SL}(f_\theta, \mathcal{D}_k^S) = \arg\min_\theta \frac{1}{n_k^s} \sum_{i=1}^{n_k^s} \mathcal{L}_{ce}(f_\theta(\boldsymbol{x}_{k,i}^s), \boldsymbol{y}_{k,i}), \tag{1}$$

where $\mathcal{L}_{ce}$ denotes the cross-entropy loss. Because the target and source domains share the same semantic classes, the source-trained model can be directly used on the target. However, its performance will degrade on the target domain because of the distributional differences between the source domains and the target domain, i.e., because $\mathcal{S}_k \neq \mathcal{T}$. The goal in single-source UDA is to leverage the target domain unannotated dataset and the source-trained model and adapt the model to have an enhanced generalization on the target domain. The common strategy for this purpose is to map the data points from the source and the target domains into a shared embedding space in which distributional differences are minimized. To model this process, we consider that the base model $f_\theta$ can be decomposed into an encoder subnetwork $g_{\boldsymbol{u}}(\cdot) : \mathbb{R}^{W \times H \times C} \to \mathbb{R}^{d_Z}$ and a classifier subnetwork $h_{\boldsymbol{v}}(\cdot) : \mathbb{R}^{d_Z} \to \mathbb{R}^{|\mathcal{Y}|}$ with learnable parameters $\boldsymbol{u}$ and $\boldsymbol{v}$, where $f(\cdot) = (h \circ g)(\cdot)$ and $\theta = (\boldsymbol{u}, \boldsymbol{v})$. In this formulation, the output-space of the encoder subnetwork models a latent embedding space with dimension $d_Z$. In a single-source UDA setting, we select a distributional discrepancy metric $D(\cdot, \cdot)$ to define a cross-domain loss function and train the encoder by minimizing the selected metric. As a result, the distributions of both domains become similar in the latent space and hence the source-trained classifier subnetwork $h_k(\cdot)$ will generalize on the target domain $\mathcal{T}$. Many UDA methods have been developed using this approach and we base our method for multi-source UDA on this solution for each of the source domains.

To address a multi-source UDA setting, a naive solution is to gather data from all source domains centrally and create a single global source dataset, and then use single-source UDA. However, this approach is not practical in medical domains due to strict regulations and concerns about data privacy and security which prevent sharing data across different source domains. Even if data sharing were permitted, another major

challenge arises from the significant differences in data distributions and characteristics among source domains. These differences can lead to negative knowledge transfer across the domains Wang et al. (2019). Negative knowledge transfer can occur because information from some source domains might be irrelevant or even harmful to the performance on the target domain. Additionally, the data from different source domains could interfere with each other, further complicating the learning process. These issues create a situation where finding a common representation that works effectively for all the source domains becomes challenging. To address these challenges, our approach of choice is ensemble learning. Ensemble learning involves combining multiple models or learners to improve overall collective performance. In the context of multi-source UDA, the idea is to develop individual single-source UDA models using each source domain and then leverage the strengths of these models and combine their predictions to make a final prediction.

## 4  Proposed Multi-Source UDA Algorithm

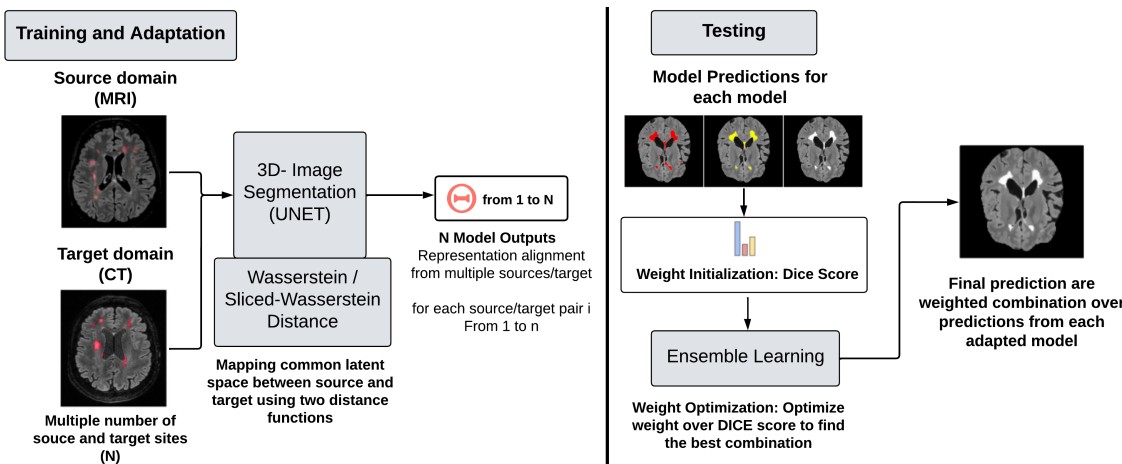

Figure 1: Block-diagram of the proposed multi-UDA approach: (a) we train source-specific models for each source domain based on ERM. (b) we perform single-source UDA for adapting each source-trained model via distributional alignment in the shared embedding space (c) we aggregate the individual source-trained model predictions to make the final prediction on the target domain predictions according to their reliability.

As illustrated in Figure 1, we follow a two-stage procedure to address multi-source UDA with MRI data. We first solve $N$ single-source UDA problems, each for one of the source domains. We then benefit from an ensemble of these distinct models. To align the target distribution with a source domain distribution, we use the Sliced Wasserstein Distance (SWD) because it a suitable metric for deep learning optimization. Because SWD has the nice property of having non-vanishing gradients Rabin et al. (2011). Moreover, SWD can be computed using a closed-form solution from the empirical samples of two distributions:

$$W_2(g(T), g(S_k)) = \frac{1}{L} \sum_{l=1}^{L} |\langle g(x_{i_l}^t), \phi_l \rangle - \langle g(x_{k,i_l}^t), \phi_l \rangle|^2 \tag{2}$$

where $\phi_l \in \mathbb{S}^{d_z-1}$ is a 1D projection direction which is drawn uniformly in a random manner from the unit ball $\mathbb{S}^{d_z-1}$. Also, $i_l$ and $j_l$ denote the sorted indices of $\{g(x_i) \cdot \phi_l\}_{i=1}^{M}$ for the source and the target domains.

We then solve the following optimization problem to adapt the model obtained from solving Eq. (1):

$$\min_{\theta} \mathcal{L}_{SL}(f_\theta, \mathcal{D}_k^S) + \gamma W_2(g_u(\mathcal{D}_k^S), g_u(\mathcal{D}^T)) \tag{3}$$

where $\gamma$ is a regularization parameter. The first term enforces the embedding space to remain discriminative and the second term aligns the two distributions in the embedding space.

After completing the adaptation process for each source domain, each model can generate a distinct mask on the target domain images. The key question in multi-source UDA is to obtain a solution that is better than these single-source UDA solutions. We obtain the final model predictions for the target domain by combining the probabilistic predictions from all $N$ adapted models. We combine the model predictions in a pixel-wise manner $\sum_{i=1}^{n} w_i f_{\theta_i}$ using mixing weights $\boldsymbol{w} = (w_1, w_2, \ldots, w_n)$, where $0 \geq w_i \geq 1$ and $f_{\theta_i}$ represents the adapted model corresponding to the $i^{th}$ source domain. This aggregation process allows for benefiting from the source models without sharing data across the source domains. Choosing the appropriate weight values is the key remaining challenge. We need to assign the weights such that the models that do not generalize well could not adversely impact the quality of the aggregated segmentation mask. To address this concern, we employ the concept of *prediction confidence* of the source model on the target domain as a proxy for the model generalization capability. To this end, we evaluate how confident the source model is when making predictions on the target domain and consider the measured confidence in the aggregation process. Intuitively, we reduce the contribution of less certain predictions. The rationale behind using prediction confidence as a basis for weight assignment is supported by empirical evidence, which we have presented in Section 5. We set a confidence threshold denoted as $\lambda$, tuned empirically, and compute the weight as follows:

$$\tilde{w}_k \sim \sum_{i=1}^{n^t} \mathbb{1}(\max \tilde{f}_{\theta_k}(\boldsymbol{x}_i^t) > \lambda), \quad w_k = \tilde{w}_k / \sum \tilde{w}_k, \tag{4}$$

where $\tilde{f}(\cdot)$ denotes the model output just prior to the final SoftMax layer. This output can be considered a probability distribution which measures certainty well. If the prediction confidence of the $k^{th}$ model exceeds $\lambda$, we assign $w_k$ to be a non-zero value to incorporate the predictions from that model into the final prediction process. However, if the prediction confidence falls below the threshold, we assign $w_k$ to be zero.

Note that we maintain data privacy during the initial stages of pretraining and adaptation by ensuring that data samples are not shared between any two source domains. When we aggregate the predictions of the resulting models, we do not need the source data at all. As a result, our approach is applicable to medical domains when the source datasets are distributed across multiple entities. Our approach also allows for benefiting from new source domains as new domains become available without requiring retraining the models from scratch. To this end, we only need to solve new single-source UDA problems. We then update the normalized mixing weights to benefit from the new domain to continually enhance the segmentation accuracy. The update process is efficient and incurs negligible runtime compared to the actual model training, offering an FL solution.

---

**Algorithm 1** Federated Multi-Source Unsupervised Domain Adaptation

1: **procedure** SINGLE-SOURCE UDA($S_i$, $T$)
2:      Pre-train $f_{\theta_i}$ by minimizing the supervised learning loss on $S_i$
3:      Update $f_{\theta_i}$ on the target domain $T$ by solving Eq. (3)
4:      **return** $f_{\theta_i}$
5: **procedure** ENSEMBLE($x$, $f_{\theta_i}$, )
6:      Compute the mixing weights $w_i$ using Eq. (4)
7:      Compute the eventual segmentation mask $M$ by computing the weighted average of the single-source UDA masks $M_i$: $M \leftarrow \frac{\sum_{i=1}^{N} w_i M_i}{\sum_{i=1}^{N} w_i}$
8:      **return** $M$

---

Our proposed approach is named "Federated Multi-Source UDA" (FMUDA), presented in Algorithm 1.

## 5 Experimental Validation

### 5.1 Experimental Setup

**Datasets:** we used the following two datasets in our experiments.

**MICCAI 2016 MS lesion segmentation challenge dataset Commowick et al. (2021):** this dataset contains MRI images from patients suffering from Multiple Sclerosis in which images contain hyperintense lesions on FLAIR. The dataset incorporates images from different clinical sites, each employing a different model of MRI scanner. Each site can be naturally modeled as a domain for form a multi-source UDA setting. In our experiments, we assume that each site has contributed images from five patients for training and ten patients for testing. The dataset is divided into training and a testing image sets. Each patient's data includes

high-quality segmentation maps derived from averaging manual annotations by seven independent manual segmentation by expert radiologists. These maps present an invaluable resource for our experimentation, offering the possibility of evaluation against gold standard used in clinical settings.

**2019 CHAOS MR Dataset Kavur et al. (2021):** This dataset consists of MR and CT scans with segmentation maps for abdominal organs such as the liver, right kidney, left kidney, and spleen. In total, 20 MR scans are obtained. We split the dataset randomly into three sites (01,02, and 03) to report results in a multi-source UDA setting. The images were re-sampled to an axial view size of $256{\times}256$. The background was then cropped such that the distance between any labeled pixel and the image borders is at least 30 pixels, and scans were again resized to $256{\times}256$. Each 3D scan was normalized independently to zero mean and unit variance, and values more than three standard deviation from the mean were clipped. Data augmentation was performed on both the training MR and training CT instances using (1) random rotations of up to 20 degrees, (2) negating pixel values, (3) adding random Gaussian noise, and (4) random cropping.

**Preprocessing & Network Architecture:** To maintain the integrity of our experiments, we have strictly used the test images solely for the testing phase, ensuring they were not used in any part of the training, validation, or adaptation processes. Following the literature on the MICCAI 2016 MS lesion segmentation challenge, we subjected the raw MRI images to several preliminary pre-processing procedures prior to using them as inputs for the segmentation network for enhanced performance. The procedures for each patient included (i) denoising of MRI images using the non-local means algorithm Coupé et al. (2008), (ii) rigid registration in relation to the FLAIR modality, performed to preserve the relative distance between every pair of points from the patient's anatomy to achieve correspondence, (iii) skull-stripping to remove the skull and non-brain tissues from the MRI images that are irrelevant to the task, and (iv) bias correction to reduce variance across the image. To accomplish these steps, we utilized Anima [1], a publicly accessible toolkit for medical image processing developed by the Empenn research team at Inria Rennes2. We employed a 3D-UNet architecture Isensee et al. (2018) as our segmentation model (please refer to the Appendices for the detailed architecture visualization) which is an improved version of the original UNet architecture Ronneberger et al. (2015b) to benefit from spatial dependencies in all directions. To ensure uniformity across the dataset, images were resampled to share a consistent size of $128 \times 128 \times 128$. From these images, 3D patches of size $16 \times 16 \times 16$ were extracted with a patch overlap of 50%, resulting in a total of 4,096 patches per image. Although using overlapping 3D patches contain more surrounding information for a voxel which in turn is memory demanding, but training on patches containing lesions allowed us to reduce training time because the inputs become smaller while simultaneously addressing the issue of class imbalance.

**Evaluation:** Following the literature, we used the DICE score to measure the similarity between the generated results and the provided ground truth masks. It is a full reference measured defined as $\frac{2 \cdot |X \cap Y|}{|X| + |Y|}$, where $X$ and $Y$ are the segmentation masks of the predicted and ground truth images, respectively. The DICE score ranges from 0 to 1, where a score of 1 indicates perfect overlap and 0 signifies no overlap. This metric is particularly suitable for evaluating segmentation tasks, as it quantifies how well the segmented regions match the ground truth, accounting for both false positives and false negative scenarios. We repeated our experiments five times and reported the average performance.

**Baselines for Comparison:** There are not many prior works in the literature on the problem we explored. To provide a comprehensive evaluation of the proposed method and measure its competitiveness, we have set up a series of comparative baselines. These baselines have been selected not only to represent standard and popular strategies in image adaptation and prediction but also to highlight the uniqueness and advantages of our approach. Additionally, some of these baselines serve as ablative experiments that demonstrate all components of our algorithm are important for optimal performance. We use four baselines to compare with our methods: (i) **Source-Trained Model (SUDA)**: It represents the performance of the best trained model using single-source UDA for target domain. This baselines serves as an ablative experiment because improvements over this baseline demonstrate the effectiveness of using multi-source UDA. (ii) **Popular Voting (PV)**: It represents assigning the label for each pixel based on the majority votes of the individual single-source adapted models. When the votes are equal, we assign the label randomly. Majority voting

---

[1]https://anima.irisa.fr/

considers all the models to be used equally. Improvements over this baseline demonstrate the effectiveness of our ensemble technique because it is the simplest idea that comes to our mind. (iii) **Averaging (AV)**: Under this baseline, prediction image results from taking the average prediction of the single-source adapted models. This method can be particularly useful when the predictions are continuous or when there's the same amount of uncertainty in individual model predictions. This baseline can also serve as an ablative experiments because improvements over this baseline demonstrate that treating all source domains equally and using uniform combination weights is not an optimal strategy (iv) **SegJDOT** Ackaouy et al. (2020): to the best of our knowledge, this is the only prior comparable method in the literature that addresses multi-source UDA for semantic segmentation of MRI images. There are other multi-source UDA techniques but those methods are developed for classification tasks and adopting them for semantic segmentation is not trivial. This baseline is a multi-source UDA method which uses a different strategy to fuse information from several source domains based on re-weighting the adaptation loss for each single-source UDA problem alignment loss function and tuning the weights for optimal multi-source performance. A benefit that our approach offers compared to SegJDOT is that we do not need simultaneous access to all source domain data.

## 5.2 Comparative and Ablative Experiments

| Method | → 07 |
|--------|------|
| SUDA | 0.199 |
| PV | 0.022 |
| AV | 0.103 |
| SegJDOT | 0.315 |
| FMUDA | 0.407 |

| Method | → 08 |
|--------|------|
| SUDA | 0.249 |
| PV | 0.152 |
| AV | 0.068 |
| SegJDOT | 0.418 |
| FMUDA | 0.395 |

| Method | → 01 |
|--------|------|
| SUDA | 0.101 |
| PV | 0.017 |
| AV | 0.029 |
| SegJDOT | 0.385 |
| FMUDA | 0.411 |

(a) Source 1 & Source 8 (b) Source 1 & Source 7 (c) Source 7 & Source 8

Table 1: Performance comparison (in terms of DICE metric) for multi-source UDA problems defined on the MICCAI 2016 MS lesion segmentation challenge dataset.

Table 1 provides an overview of our comparative results. We have provided results for all the three possible multi-source UDA problems, wherein each instance involves designating two domains as source domains and the third domain in the dataset as the target domain. We report the downstream performance on the target domain for each UDA problem in Table 1. We have followed the original dataset to use "01", "07", and "08" to refer to the domains (sources) in the dataset. Upon careful examination, it is evident FMUDA stands out by delivering state-of-the-art (SOTA) performance across all the three multi-source UDA tasks. Particularly, improvements over SUDA is significant which demonstrate the advantage of our approach. A notable finding is also the substantial performance gap between FMUDA and PV or AV. This discrepancy serves as compelling evidence for the effectiveness and indispensability of our ensemble approach in ensuring superior model performance. It emphasizes that the careful integration of information from multiple source domains, as facilitated by FMUDA, contributes significantly to overall multi-domain UDA successful strategy. The comparison between PV and AV against SUDA reveals that multi-source UDA is not inherently a superior method when aggregation is not executed properly. PV and AV exhibit underperformance in comparison to SUDA, emphasizing the importance of a well-crafted aggregation strategy in realizing the potential benefits of multi-source UDA to mitigate the effect of negative knowledge transfer. Underperformance compared SUDA suggests that interference between source domains is a major challenge that needs to be addressed in multi-source UDA. SegJDOT addresses this challenge and exhibits a better performance but not as good as FMUDA. We think this superiority stems from the fact that FMUDA uses distinct models for each source domain. In summary, our findings suggest that our FMUDA is not only a competitive method but also compares favorably against alternative methods.

Table 2 provides comparative results using the 2019 CHAOS Grad Challenge dataset, where we tested our method in a multi-source UDA setting. We observe that our approach outperforms the other baselines with a considerable margin. The results emphasized that in the multi-source situation, our ensemble method is able to improve the image segmentation performance using the knowledge trained in multiple models.

| Method | → 03 |
|--------|------|
| SUDA   | 0.523 |
| PV     | 0.159 |
| AV     | 0.310 |
| FMUDA  | 0.588 |

| Method | → 02 |
|--------|------|
| SUDA   | 0.579 |
| PV     | 0.192 |
| AV     | 0.328 |
| FMUDA  | 0.615 |

| Method | → 01 |
|--------|------|
| SUDA   | 0.543 |
| PV     | 0.253 |
| AV     | 0.377 |
| FMUDA  | 0.602 |

(a) Source 1 & Source 2   (b) Source 1 & Source 3   (c) Source 2 & Source 3

Table 2: Performance comparison (in terms of DICE metric) for multi-source UDA problems defined on the 2019 CHAOS Grad challenge dataset (CHAOS MR).

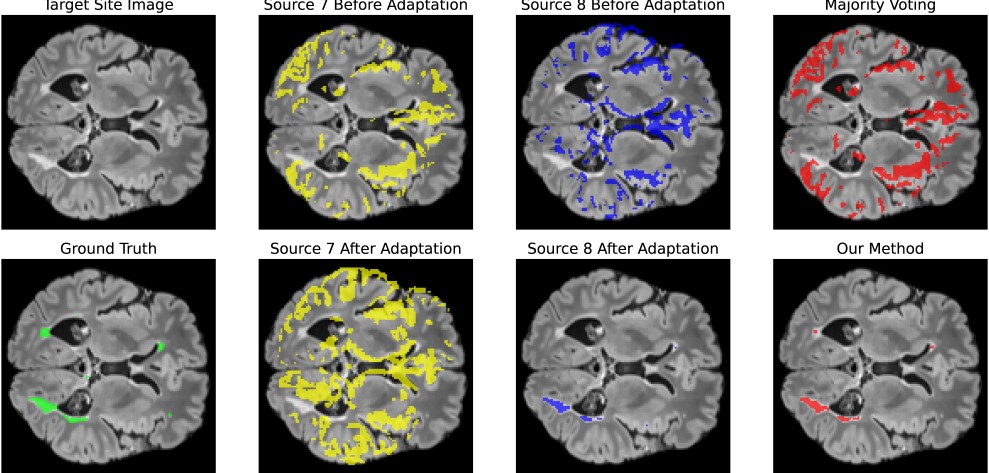

Figure 2: Segmentation masks generated for a sample MRI image when Source "01" is used as the source domain in UDA. In each figure, the colored area shows the mask generated by each UDA model.

To offer a more intuitive comparison and provide a deeper insight about the comparative experiments, Figure 2 showcases segmentation results along with the original segmentation mask of radiologists when Source "01" is served as the target domain and Sources "07" and "08" are used as the UDA source domains. Through inspecting the second and the third columns, we note that the performance of the single-source UDA methods is quite different. While source "08" leads to a decent performance, source "07" does not lead to a good UDA performance. This observation is not surprising because UDA is effective when the source and the target domain share distributional similarities and this example suggests that source "07" is not a good source domain to perform segmentation in on source "01". We can understand why the best single-source UDA method can have a better performance. Additionally, this example demonstrates that as opposed to intuition, using more source domains does not necessarily lead to improved UDA performance due to the possibility of negative knowledge transfer across the domains. In situations in which the source domains are diverse, aggregation techniques such as averaging or majority vote are not going to be very effective because unintentionally we will give a high contribution to the source domains with low-performance when generating the aggregated mask. Hence, it is possible that the aggregated performance is dominated by the worse single-UDA performance. It is even possible to have a performance less than all single-source UDA models when individual single-source UDA domain models lead to inconsistent predictions. Note that majority voting also can fail because the majority of the models can potentially be low-confidence models. In other words, multi-source UDA should be performed such that good source domains contribute the most when the aggregated mask is generated. In the absence of such a strategy, the multi-source UDA performance can even lead to a lower performance than single-source UDA. The strength of FMUDA is that, as it can be seen in Figure 2, it can aggregate the generated single-source UDA masks such that the aggregated mask would become better than the mask generated by each of the single-source UDA models. For example, although Source "08" model leads to a relatively good performance, it misses to segment two regions in the upper-half of

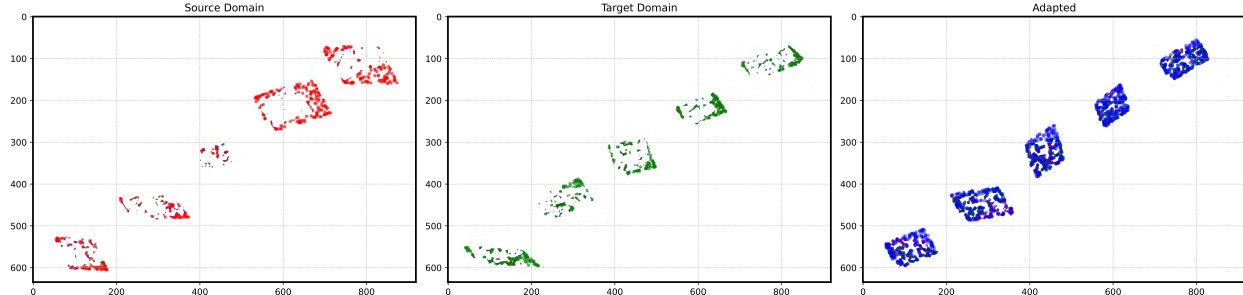

Figure 3: Distribution matching in the embedding space: we use UMAP for visualization of data representations when Source "07" in the dataset is used as the UDA source domain and Source "01" of the dataset is used as the UDA target domain: (Left) source domain; (Center) target domain prior to single-source model adaptation; and (Right) target domain after single-source model adaptation

the brain image. The multi-source UDA model, can at least partially include these regions using the "07" model. This improvement stems from using the "8" domain which is confident on those regions.

To offer an intuitive insight about the way that our approach works, Figure 3 illustrates the affect of domain alignment on the geometry of the data representations in the shared embedding space. In this figure, we have reduced the dimension of data representations in the shared embedding space using UMAP tool McInnes et al. (2018) to two for visualization purpose. In this figure, we showcase the latent embeddings of data points for the source domain (Source "08" in the dataset) and the target domain (Source "01" in the dataset) both before and after adaptation to study the impact of single-source UDA on the geometry of data representations. Each point in the figure corresponds to a pixel. Through careful visual inspection, we see that FMUDA effectively minimizes the distance between the empirical distributions of the target domain and the source domain after adaptation, leading to learning a domain-agnostic embedding space at the output-space of the encoder. Although the eventual mask is generated by aggregating several models, alignment of single-source UDA distribution pairs can translate into an enhanced collective performance because each model become more confident after performing single-source UDA. This experiment highlights the efficacy of FMUDA in facilitating domain adaptation and improving the overall performance across diverse domains.

In addition to the exploration of multi-source UDA setting, we conducted single-domain UDA experiments and compared our results against SegJDOT, showcasing the competitiveness of our proposed approach in this scenario. The results of these experiments are summarized in Table 3, where we present performance results for six distinct pairwise single-source UDA problems defined on the dataset. To ensure a fair evaluation, we aligned the training/testing pairs with those used in SegJDOT. The observation from the results is that our proposed approach consistently outperformed SegJDOT. Notably, when considering the average DICE score across these tasks, our approach exhibited a remarkable $\approx 20\%$ improvement over the SegJDOT. This heightened performance is because SegJDOT relies on optimal transport for domain alignment, but our approach leverages SWD for distribution alignment. The inherent characteristics of SWD contribute to the improved adaptability and effectiveness of our method. This experiment demonstrate a second angle of our novelty in using SWD for solving UDA for semantic segmentation. Our proposed is a competitive method for single-source UDA for problems involving semantic segmentation. These results indicate that our improved performance in the case of multi-source UDA also stems from performing single-source UDA better.

In Table 4, we present results for a single-source UDA scenario where we have reported results on the CHAOS dataset. Our decision to report results in this setting is motivated by the fact that previous studies have primarily reported their performances in a single-source UDA context. We compare our performance against single-source UDA methods, including, SIFA Chen et al. (2019), CyCADA Hoffman et al. (2018), CycleGAN Zhu et al. (2017), SynSeg-Net Huo et al. (2018), and AdaOutput Tsai et al. (2018). Our results show that FMUDA achieves SOTA performance in this setting. We note that achieving SOTA performance in this context is also beneficial for multi-source UDA scenarios. This is because the overall SOTA performance in multi-source UDA depends on the individual models' SOTA performance.

**(a) Source 1**

| Method | → 07 | → 08 | Avg. |
|---|---|---|---|
| Pre-Adapt | 0.090 | 0.430 | 0.260 |
| SegJDOT | 0.110 | 0.470 | 0.290 |
| FMUDA | 0.452 | 0.418 | 0.435 |

**(b) Source 7**

| Method | → 01 | → 08 | Avg. |
|---|---|---|---|
| Pre-Adapt | 0.430 | 0.390 | 0.410 |
| SegJDOT | 0.450 | 0.440 | 0.445 |
| FMUDA | 0.484 | 0.442 | 0.463 |

**(c) Source 8**

| Method | → 01 | → 07 | Avg. |
|---|---|---|---|
| Pre-Adapt | 0.350 | 0.070 | 0.210 |
| SegJDOT | 0.450 | 0.290 | 0.370 |
| FMUDA | 0.483 | 0.458 | 0.471 |

Table 3: Performance comparison (in terms of DICE metric) for single-source UDA tasks defined on the MICCAI 2016 MS lesion segmentation challenge dataset.

| Method | Dice | | | | |
|---|---|---|---|---|---|
| | Liver | R.Kidney | L.Kidney | Spleen | Average |
| Source-Only | 73.1 | 47.3 | 57.3 | 55.1 | 58.2 |
| Supervised | 94.2 | 87.2 | 88.9 | 89.1 | 89.8 |
| SynSeg-Net | 85.0 | 82.1 | 72.7 | 81.0 | 80.2 |
| AdaOutput | 85.4 | 79.7 | 79.7 | 81.7 | 81.6 |
| CycleGAN | 83.4 | 79.3 | 79.4 | 77.3 | 79.9 |
| CyCADA | 84.5 | 78.6 | 80.3 | 76.9 | 80.1 |
| SIFA | 88.0 | 83.3 | 80.9 | 82.6 | 83.7 |
| FMUDA | 89.1 | 72.5 | 81.4 | 80.4 | 80.9 |

Table 4: Performance comparison (in terms of DICE metric) for single-source UDA problems defined on the 2019 CHAOS Grad challenge dataset (CHAOS MR)

## 5.3 Analytic Experiments

In Figure 4, we first study the dynamics of our adaptation strategy on the model performance under the utilization of Source "01" of MICCAI 2016 MS lesion segmentation dataset as the source domain. In this figure, we have visualized the training loss and the target domain performance versus training epochs. We observe a consistent pattern in both domains: pre-training on the source domain consistently enhances performance in the target domains due to cross-domain similarities. Furthermore, a notable uptick in target domain accuracies becomes evident as the adaptation process initiates.

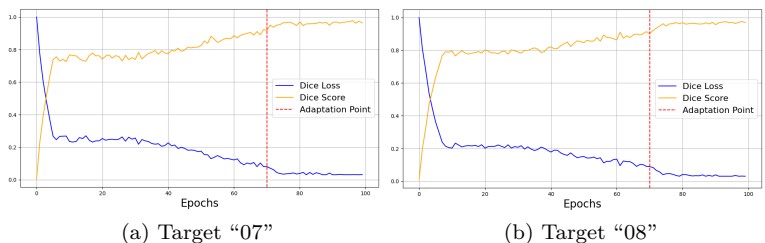

(a) Target "07"          (b) Target "08"

Figure 4: Effect of the pretraining and adaptation process on the target domain performance (yellow curve) and the training loss (blue curve) for Source "01" of the MICCAL 2016 dataset.

Finally, we studied the sensitivity of our performance with respect to major hyperparameters that we have using the MICCAI 2916 MS lesion segmentation dataset. We study the effect of the value of confidence parameter *lambda* on the downstream performance. This parameters acts as a threshold to filter out noises in images from multiple site. To this end, we have measured the model performance versus the value of $\lambda$ on each target domain. Figure 5 presents the results for this study. We observe that the value for this parameter is important and selecting it properly is very important. Based on the observations, we conclude $\lambda = 0.3$ is a suitable initial value for this parameter in our experiments. We can also use the validation set to tune this parameter for an optimal performance.

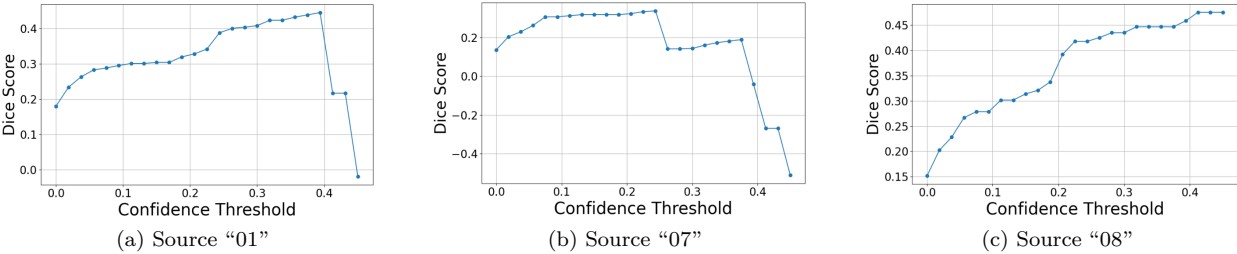

(a) Source "01"        (b) Source "07"        (c) Source "08"

Figure 5: Mode Performance versus the value for the hyperparameter $\lambda$.

We also investigate the influence of the SWD projection hyper-parameter, denoted as $L$ in definition of SWD in Equation 2. While a larger value of $L$ results in a more precise approximation of the SWD metric, it also comes with the drawback of increased computational load to compute SWD. Our objective is to determine whether there exists a range of $L$ values that provides satisfactory adaptation performance and to scrutinize the impact of this parameter. To this end, we use two UDA tasks, as illustrated in Figure 6. We present our findings based on a range of $L$ values $L \in 1, 25, 50, 100, 150, 200, 250$. As anticipated, tightening the SWD approximation by increasing the number of projections results in improved performance. However, we observe that beyond a certain threshold, approximately when $L \approx 50$, the performance gains become marginal and the algorithm becomes almost insensitive. Consequently, $L = 50$ is a good choice for this particular hyper-parameter to balance the computational efficiency and adaptation performance.

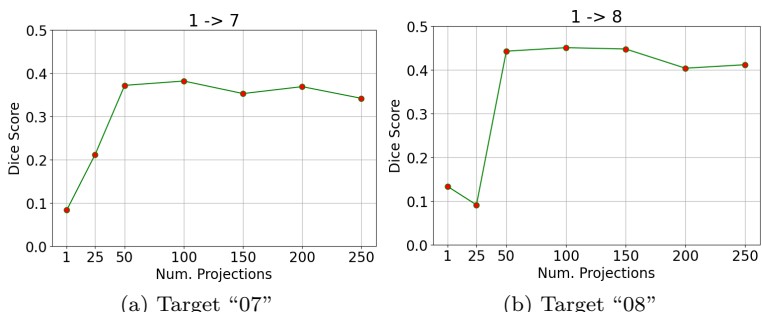

(a) Target "07"        (b) Target "08"

Figure 6: Performance in target domain versus the number of projections used in computing SWD.

## 6 Conclusion

We developed a multi-source UDA method for segmentation of medical images, when the source domain images are distributed. Our algorithm is a two-stage algorithm. In the first stage, we use SWD metric to match the distributions of the source and the target domain in a shared embedding space modeled as the output of a shared encoder. As a result, we will have one adapted model per each target-source domain pair. In the second stage, the segmentation masks generated by these models are aggregated based on the reliability of each model to build a final segmentation map that is more accurate than all the individually generated single-source UDA masks. The validity of our algorithm is supported by experimental results on two real-world medical image datasets. Our experiments showcase the competitive performance of our algorithm when compared to SOTA alternatives. Our algorithm also maintains data privacy across the source domains because source domains do not share data. Future endeavors involve exploring scenarios where the data for source domains is fully private and cannot be shared with the target domain. Another limitation of our work is heuristic nature of aggregation which despite being practical, may require further theoretical analysis. We also lack a theoretical approach to tune the hyperparameters which would be beneficial when validation data is lacking and empirical hyperparameter tuning is not feasible.

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

## A Appendix

### A.1 Optimal Transport for Domain Adaptation

Optimal Transport (OT) is a probability metric based on determining an optimal transportation of probability mass between two distributions. Given two probability distributions $\mu$ and $\nu$ over domains $X$ and $Y$, OT is defined as:

$$W(\mu, \nu) = \inf_{\gamma \in \Pi(\mu, \nu)} \int_{X \times Y} d(x, y) d\gamma(x, y) \tag{5}$$

where $\Pi(\mu, \nu)$ represents the set of all joint distributions $\gamma(x, y)$ with marginals $\mu$ and $\nu$ on $X$ and $Y$, respectively. The transportation cost is denoted as $d(\cdot, \cdot)$, which can vary based on the specific application. For instance, in many UDA methods, the Euclidean distance is used. However, computing the OT involves solving a complex optimization problem and can be computationally burdensome. Alternatively, SWD reduces the computational complexity while retaining the foundational benefits of OT.

### A.2 The Segmentation Architecture

Figure 7 presents the architecture of the 3D U-Net that we used in our experiments.

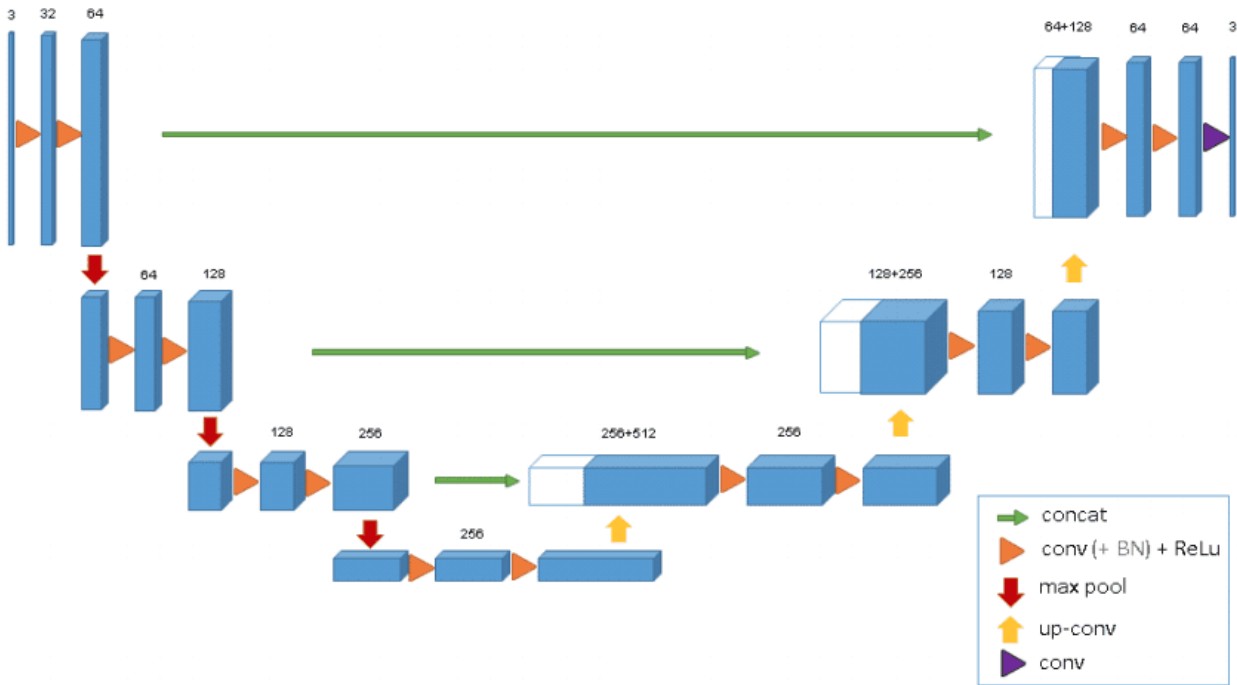

Figure 7: 3D-UNET architecture

### A.3 Details of Setting the Optimization Method

We used ADAM because it is well-suited for problems that are large in scale and have sparse gradients. It combines the advantages of both Adaptive Gradient Algorithm (AdaGrad) and Root Mean Square Propagation (RMSProp), allowing it to handle non-stationary objectives and noisy gradients.

- **Initialization:** We initialized the weights to be optimized and set hyperparameters such as the learning rate, first and second moment estimates, and smoothing terms according to common best practices.

- **Iteration:** During each iteration, the optimizer computes the gradients of the DICE score with respect to the weights, and updates the weights in a direction that is expected to increase the DICE score.

- **Adaptive Learning Rates:** ADAM dynamically adjusts the learning rates during optimization, using both momentum (moving average of the gradient) and variance scaling. This makes it robust to changes in the landscape of the objective function.

- **Termination Criteria:** The optimization was terminated upon convergence, which could be determined by a number of epochs or a tolerable change in the DICE score.

