# OpenReview forum: "Unsupervised Federated Domain Adaptation for Segmentation of MRI Images"
_TMLR — Rejected by TMLR_

### Review · Reviewer_pBzx · 2024-01-19

**Summary Of Contributions:**

The paper presents an unsupervised federated domain adaptation approach for MRI segmentation. The approach is composed of two stages, first runs multiple unsupervised domain adaptations, and then enables the outputs of all models. The method is validated on one dataset (MS lesion) and is showed to outperform four prior works on this dataset.

**Audience:**

Yes

**Claims And Evidence:**

No

**Requested Changes:**

The presentation of the paper is not easy to get through. I read the introduction and methodology several times to get an idea about paper’s contribution and model design. Below are some suggestions w.r.t. paper presentation.

-	Figure 1 is unclear. Improving the presentation of this figure would ease understanding of the method.

-	The reviewer finds missing a proper positioning of the methodology to prior art. From the methodology section, it is unclear which parts of the method already existed (e.g. were introduced in prior works) and which are new. Introducing a Background section that would cover elements of the method previously introduced would strengthen the presentation.

-	From the introduction, it is unclear what are the contributions that the paper is claiming. Listing the contributions in the introduction would benefit the manuscript.

* One claim that is a bit unclear to the reviewer is the privacy preserving one, that no data need to be shared as all the models are run locally. However, it looks that the target data needs to be sent to all source data sites. Could the authors comment on this?

Experiments contain a fair number of ablations; however there is only one dataset used in the evaluations.

-	Why the authors use the MS lesson data if it “has not been explored extensively in UDA settings”?
-	Adding another dataset would provide more support for the claims. Would it be possible to evaluate the method on this challenge https://crossmoda2022.grand-challenge.org/?
-	The authors mention that they average multiple annotations to obtain the ground truth. This operation makes the segmentation masks not binary. How the authors deal with this?
-	It might be nice to add confidence intervals or standard deviations to Tab 1.

Limitations of the approach are not discussed. Adding a paragraph that discusses the limitations of the method would strengthen the manuscript.

Minor issues:
- Algorithm 1: Please define all the inputs to the procedures.
- “A major reason for this limitation is that training deep neural networks requires large annotated datasets.” Do the authors have a reference for this claim?
- “crowdsourcing annotation platforms would be inapplicable because medical data is normally distributed in different institutions”. Probably the privacy-related aspects are the biggest issue here.
- It is unclear to the reviewer what do the authors mean with “probability metric”.

**Strengths And Weaknesses:**

Strengths.
- Paper deals with an important problem.
- The proposed pipeline looks reasonable.
- The paper contains ablations of hyperparameters.

Weaknesses (for details see below).
- The presentation of the paper is not easy to follow.
- Some claims might require some clarifications.
- Experimental section could be strengthened.

---

> ### Author Response · Authors · 2024-02-24
> **Response to the reviewer**
>
> We thank the reviewer constructive comments. We are glad that the reviewer has found our work important and reasonable. We hope that the following changes can address your concerns:
>
> Weakness1,2,3: we improve the quality of Figure 1 to clearly present our method by adding description and mentioned the all distance functions we chose to test in our method. Moreover, we restructured the explanation of existing method we used in the experiment and our approach. The list of contribution has been added in the introduction. We are also happy to improve further if the reviewer provides us more detailed feedback.
>
> Weakness4: The reviewer is correct. What we meant by privacy is privacy between the source domains. It is of course a limitation for our work that the target domain data does not remain private but in practice, when the user wants to solve the target domain problem, privacy of its data is less critical that privacy between the sources. We have explained this limitation in our conclusion
>
> Weakness5: Although the MICCAI 2016 brain scan MS lesion dataset has not been used extensively in UDA, it is a perfect dataset for our goal because we would like to study  a multi-source DA setting and the dataset is collected from multiple sites. In other words, federated learning in medical image segmentation has not been studied extensively and we are one of the first to address this task.
>
>
> Weakness6: We agree with the reviewer that adding results on more datasets would strengthen our claim results. We tried using https://crossmoda2022.grand-challenge.org/ for this purpose, but the crossmoda datasets contain 105 MRI scan images which is too big for us to train and replicated related results during the rebuttal period. Instead, used the 2019 CHAOS Grad Challenge MR dataset, as seen section 5.2.
> Weakess7: In order to compare with the ground truth, we use soft segmentation and then thresholding to obtain the final mask, just similar to deriving a binary mask from the individual model predictions.
>
> Weakness8: Due to the limited time, we could not rerun our code multiple times during the rebuttal period to report the confidence intervals. We are running our code and if our work is accepted, we will add the confidence intervals in our results.
>
> Weaknss8: We added a short sentence about the limitations of our work in the conclusion section.
>
> Weakness9,10: we added a few references.
>
> Weakness11: A probability metric is a functional that receives samples of two probably distributions and returns a number that shows how similar they are.

---

> ### Author Response · Authors · 2024-03-04
> **Request for continual discussion**
>
> Dear Reviewer,
>
> We reiterate our appreciation for your time. We think that your concerns can be addressed and respectfully ask you to read our response and check the changes we did in the draft. If possible, we respectfully ask you to engage in discussion with us if you feel your concerns have not been addressed. Specifically, we hope you check the new experiments we did. We are hopeful that your time allows continual discussion so you can make your final recommendation when all your concerns are addressed.
>
> Best, Our team

---

> ### Author Response · Authors · 2024-03-13
> **Follow-Up Request**
>
> Dear Reviewer,
>
> We totally understand that reviewing is a volunteer activity and thank you for your time and feedback that helped improving our work. Given that we are closing to the deadline announced by the AE,  we are hopeful that you check our response and let us know whether you feel your concerns have been addressed? If not, we respectfully ask you to let us know so we use the time to address your concerns as much as we can so you make your recommendations when all your concerns are addressed.
>
> Kind regards,
>
> Our team

---

### Review · Reviewer_yhZk · 2024-01-25

**Summary Of Contributions:**

This manuscript proposes a method for federated multi-source domain adaptation for MRI segmentation. The approach is multi-source because it exploits multiple different domains to learn from MRI data coming for instance from different machines and/or hospitals. It is also federated because it learns each source domain independently, such that the source data does not need to move.
The approach trains independent target-adapted models based on the Sliced Wasserstein Distance. Then the pixel-wise prediction of those models are combined with weights that are computed proportionally to the number of pixels with maximum confidence that is higher than a given threshold $\lambda$.
The presented results show that the proposed method outperforms some baselines and another approach on a MRI dataset comprised of 3 different domains.

**Audience:**

Yes

**Broader Impact Concerns:**

I do not see any specific ethical issues or implications.

**Claims And Evidence:**

No

**Requested Changes:**

- Clarify how and when the DICE score is used for the estimation of the weights associated to the ensemble.

- Add related work about federated learning and model ensembling.

- Add experiments on more datasets and more domains to validate the results.

- Clarify how the model's hyper-parameters are tuned. It is fair to compare their model with baselines that do not require any hyper-parameter?

- Explain their algorithm and add more details on how the baselines are implemented and why they perform so poorly.

- Correct the typos in the formulation and text.

**Strengths And Weaknesses:**

\+ The idea of federated adaptation makes a lot of sense especially in the medical imaging field.

\+ The paper is in general well written and quite clear.

\- The proposed idea is more an application of existing techniques in my opinion and it could better fit an application-oriented journal.

\- There is no related work on ensembling and federated learning, which are two important building blocks of the proposed approach

\- Algorithm 1 is not explained and some parts are not defined. For instance, what is DICE(M)? It seems that the weights $w_i$ for the ensemble are estimated based on that , while in the text authors present eq. 4. Also the use of DICE score seems to be used also in Fig 1. DICE means that the DICE score of a given segmentation mask is computed? This would mean using annotations on target data, which is not valid.

\- The experimental evaluation seems limited. Only 1 dataset with 3 domains, two as source and one for target. Would the method work on a different dataset? Would the method work with more source domains?

\- The comparison with only another method is limited. Although there are not other multi-source approaches for segmentation, the adaptation of standard approaches to segmentation seems feasible and would make the authors' statements stronger.

\- In ensembling, averaging the probabilities of the different models is a strong baseline. Here instead it performs quite poorly. Could the authors investigate the reason? In this baseline the authors average the logits or the normalized probabilities of the models?

\- Based on section 6.3 it seems that the hyper-parameters of the proposed approach ($\lambda$ and $L$) are optimized on the target data. Is it a valid assumption? At the beginning of page 12 authors state that $\lambda$ can also be optimized on a validation set. However, in the given split there is no validation set. Furthermore, the problem is unsupervised domain adaptation, thus, also the validation data, if available should be unlabelled.

\- The proposed formulation contains some typos. e.g. eq. 2 does not close a parenthesis, eq. 3 $u_u$ instead of $ g_u$ and missing $\gamma$.

\- Some sentences are unfinished or some words are missing.

\- The theoretical analysis does not add much to the paper as it is a simple adaptation of Redko & Sebban (2017).

---

> ### Author Response · Authors · 2024-02-24
> **Response to the reviewer**
>
> We thank the reviewer for detailed comments. We are glad to see that you have identified the strength of our paper. We have tried to address your concern and are hopeful that the updated manuscript satisfies your expectations about requested changes. Please see check the changes we have done according to your request:
>
> Weakness1: we would like to respectfully request that the reviewer checks the following TMLR manuscript as a subset of practical papers in TMLR:
>
> Revisiting Hidden Representations in Transfer Learning for Medical Imaging
>
> PCPs: Patient Cardiac Prototypes to Probe AI-based Medical Diagnoses, Distill Datasets, and Retrieve Patients
> Self-supervised Learning for Segmentation and Quantification of Dopamine Neurons in Parkinson’s Disease
>
> As it can be seen, TMLR has a history of publishing practical works and in this sense our work is not very different from the above.
>
> Weakness2: We added a paragraph about federated learning in the related work section. Please let us know in case we are missing any related work.
>
>
> Weakness3: Thank you for pointing out this inconsistency. Algorithm 1 presentation is incorrect and we have missed to update Algorithm 1 after we had our final version. We corrected Algorithm 1 and made it consistent with the text.
> Weakness4: In response to this shared concern with the other reviewers, we have performed new experiments using a second dataset. Please check section 5.2.
>
> Weakness5: To address this concern, we offered comparison of our method with UDA methods on the new dataset in Table 4. We used single-source UDA because most previous methods consider this setting and outperforming them demonstrate that our method will outperform them in multi-source UDA because of better performance in the individual domains.
>
> Weakness6: We speculate that the reason is that the probabilities of the different models are not indeed trustworthy at the same level. It is well known that neural networks are overconfident about their prediction which may not be the result. As a result, averaging the models may not lead to an optimal performance.
>
> Weakness7: This is an excellent point. As you said, our problem is that we don’t have a validation set to tune these hyperparameters. If a dataset provides a validation set, we can easily tune these hyper parameters. However, Figure 5 and 6 demonstrate that our performance is not extremely sensitive to the hyperparameter values. As seen in figure 6, it only suffices to set l more than 50 to have a decent performance. Figure 5 shows that performance with respect to $\lambda$ is more sensitive but the value of 0.3 that we used works relatively well.
>
> Weakness8: Thank you for reading the paper carefully. We corrected the typos.
>
> Weakness9: We did a pass and tried to improve the presentation.
>
> Weakness10: In response to this concern as well as increased length of the paper due to new content, we fully removed the theoretical section.

---

> ### Author Response · Authors · 2024-03-04
> **Request for continual disucssion**
>
> Dear Reviewer,
>
> We reiterate our appreciation for your time. We think that your concerns can be addressed and respectfully ask you to read our response and check the changes we did in the draft. If possible, we respectfully ask you to engage in discussion with us if you feel your concerns have not been addressed. Specifically, we hope you check the new experiments we did. We are hopeful that your time allows continual discussion so you can make your final recommendation when all your concerns are addressed.
>
> Best, Our team

---

> ### Author Response · Authors · 2024-03-13
> **Follow-Up Request**
>
> Dear Reviewer,
>
> We totally understand that reviewing is a volunteer activity and thank you for your time and feedback that helped improving our work. Given that we are closing to the deadline announced by the AE,  we are hopeful that you check our response and let us know whether you feel your concerns have been addressed? If not, we respectfully ask you to let us know so we use the time to address your concerns as much as we can so you make your recommendations when all your concerns are addressed.
>
> Kind regards,
>
> Our team

---

### Review · Reviewer_oYEq · 2024-02-06

**Summary Of Contributions:**

The paper proposes an unsupervised domain adaptation algorithm to improve the lesion segmentation performance on the target dataset. The algorithm adds a regularization term of the distance between source and target embedding distributions. It also ensemble the outputs of models trained from multiple sources.

**Audience:**

Yes

**Broader Impact Concerns:**

No impact concerns.

**Claims And Evidence:**

No

**Requested Changes:**

Please clarify the ambiguous definitions and include additional experimental results.

**Strengths And Weaknesses:**

Strengths:

- The problem of UDA is well-motivated on medical datasets. The paper proposes a straightforward solution to improve the performance.

Weakness:

- The equations are not clearly defined. In Eq(2), how are $l$, $x_{i_l}^t$ and $x_{k, i_l}^t$ defined? There are only $T$ and $S_k$ on the LHS, but there are samples on the RHS. How are they sampled during training? In Eq (3), $u_u$ seems like a typo. Should it be $g_u$ instead? Also, where is $\gamma$ mentioned afterward?

- The theoretical analysis is not well-motivated. Theorem 1 does not give an additional message than Eq (3), since it does not show how tight the upper bound is. Also, $w_k$ is not defined here.

- The performances on only part of the domains are shown.

- Additional ablation studies are required to show which part of the model (ensembling or new loss function) contributes the most to the improvement.

---

> ### Author Response · Authors · 2024-02-24
> **Response to the reviewers**
>
> We thank the reviewer for the time and efforts. We are glad that the reviewer has found our problem well-defined and our solution sound. We think the weaknesses raised by the reviewer are easy fixes and we hope our responses below are convincing:
>
> Weakness1: Thank you for pointing out the unclarity. We improved the explanation after Eq. 2.
>
> Weakness2: After updating the manuscript with new experiments, we could not fit the manuscript in 12 pages, as required by TMLR. To handle this issue and to address the reviewers concern, we fully removed the theoretical analysis in the updated manuscript. We tend to agree that by not including the theoretical analysis, the message of the paper is not harmed.
>
>
> Weakness3 and Weakness4: In response to these weaknesses and the requested change, we performed new experiments on a new dataset. We hope that reporting results on this new dataset can address this concern. We added more ablation study results so as to, as the reviewer suggested, show that the proposed ensemble method contributes for the improved performance. We included the new results in section 5.2. We demonstrate that our proposed method works well in other medical image domains.

---

> > ### Author Response · Authors · 2024-03-04
> > **Request for continuing the discussion**
> >
> > Dear Reviewer,
> >
> > We reiterate our appreciation for your time. We think that your concerns can be addressed and respectfully ask you to read our response and check the changes we did in the draft. If possible, we respectfully ask you to engage in discussion with us if you feel your concerns have not been addressed. Specifically, we hope you check the new experiments we did. We are hopeful that your time allows continual discussion so you can make your final recommendation when all your concerns are addressed.
> >
> > Best, Our team

---

> ### Author Response · Authors · 2024-03-13
> **Follow-Up request**
>
> Dear Reviewer,
>
> We totally understand that reviewing is a volunteer activity and thank you for your time and feedback that helped improving our work. Given that we are closing to the deadline announced by the AE,  we are hopeful that you check our response and let us know whether you feel your concerns have been addressed? If not, we respectfully ask you to let us know so we use the time to address your concerns as much as we can so you make your recommendations when all your concerns are addressed.
>
> Kind regards,
>
> Our team

---

### Author Response · Authors · 2024-02-24
**General not to the reviewer**

Dear Reviewers,

We thank you for the constructive comments and feedback that helped improving our results. Other than comments about the presentation style, we did two major changes in the manuscript:

1. We perform experiments using a new dataset which led to adding two new tables to report our new results.

2. In response to two of the reviewers, we removed the theoretical section to maintain the TMLR page limit.

We think that the collective opinion of the reviewers is in the positive side and we hope that through continual engagement, we can address all the raised concerns. Please check our responses and engage in discussion if you feel more interaction can help addressing the remaining concern.

Kind regards,

Our team

---

### Decision · Action_Editor_EBrL · 2024-03-21

**Recommendation:** Reject

**Comment:**

This paper introduces an approach for unsupervised federated domain adaptation using multiple annotated source domains, where data from different sources are learned independently to protect privacy.

In the initial reviews, the reviewers raised concerns regarding paper clarity, its positioning with respect to the literature, invalidated claims, and limited experiments. During the discussion, the authors provided new experiments on an additional dataset in response to the reviewers' feedback. However, the rebuttal was not entirely convincing. The new results revealed that the proposed approach was outperformed by baseline methods, and the comparison was not reported for the first dataset. Additionally, there remained presentation issues and concerns about hyperparameter selection.

The AE carefully reviewed the submission and discussions. While acknowledging the paper's addressing of an interesting problem, the AE believes that significant improvements are necessary. The AE recommends consolidating the validation of claims, including a systematic comparison with recent baselines, analyzing the impact of hyperparameters, and specifying a clear procedure for setting them up. Therefore, the AE recommends rejection.

**Audience:**

The submission addresses the design of unsupervised domain adaptation in the federated setting, which is of interest for the TMLR audience.

**Claims And Evidence:**

The claims are not enough supported by evidence.